# LEARNING MULTI-GRANULARITY VISUAL-TEXTUAL ALIGNMENT FOR ZERO-SHOT ANOMALY DETECTION

## ABSTRACT

Utilizing a detection model trained on an auxiliary dataset to detect anomalies has shown strong potential for zero-shot anomaly detection (ZSAD). However, prior approaches typically rely on text prompts with single-level visual-textual representations, hindering the detection of anomalies that vary in shape and appearance. To address this limitation, we propose a generalized ZSAD framework that empowers visual-textual representations from coarse-grained alignment to multi-granularity alignment. On the textual side, we expand the traditional single-level alignment into a multi-level paradigm. Different from previous work that ensembles multiple prompts with limited perception, we novelly assign prompts with multiple receptive fields to facilitate the learning of structured visual semantics at different levels of granularity. This constitutes our MPA approach. Building upon MPA, we further enhance visual granularity by employing different experts for fine-grained modeling of visual patch tokens. To this end, we propose a Mixture-of-Experts adaptation mechanism that dynamically routes patch tokens to multiple experts from a shared expert pool. This allows the selected experts, each with specialized knowledge, to collaboratively represent visual tokens at multiple granularities. These components constitute our MPAMA framework. We evaluate both MPA and MPAMA on datasets across industrial and medical domains. Extensive experiments demonstrate that our method outperforms state-of-the-art approaches.

## 1 INTRODUCTION

Anomaly detection plays a vital role in various applications, including industrial inspection (Bergmann et al., 2019; Huang et al., 2022; Chen et al., 2022; Bergmann et al., 2020; You et al., 2022; Liznerski et al., 2021; Gu et al., 2024b; Gao, 2024) and medical diagnosis (Pang et al., 2021; Qin et al., 2023; Liu et al., 2023). With the advent of the pre-trained model era, anomaly detection has evolved from small models trained from scratch (Xie et al., 2023; Ding et al., 2022) to generalized architectures pre-trained on large-scale datasets (Radford et al., 2021; Zhou et al., 2022b; Jeong et al., 2023; Zhou et al., 2024a;b). The scope of detection has also broadened from category-specific designs to models capable of handling diverse objects.

Recent ZSAD approaches (Zhou et al., 2024a; Cao et al., 2024; ZHU et al., 2025; Chen et al., 2022; Jeong et al., 2023) leverage the strong generalization capabilities of large pretrained vision-language models such as CLIP (Radford et al., 2021), and have demonstrated strong potential in detecting anomalies on previously unseen objects. These methods typically employ single-level textual and visual representations to align with semantic-rich local visual features. However, they face two key limitations in Figure 1(a): (1) **Single-level alignment:** Anomalies can vary significantly in spatial scale, ranging from a tiny hole on a screw to a large crack in a hazelnut. Relying on a single-scale alignment limits the model to capturing such diverse anomaly patterns; (2) **Lack of visual granularity:** A single visual expert shared across all patch tokens restricts the model's capacity to represent anomaly patterns at varying levels of granularity.

To address these, we propose a novel multiple-granularity framework designed to empower the limited textual and visual representation for accurate anomaly semantics learning. As illustrated in Figure 1(b), we present two innovations to advance the cross-modal alignment of CLIP from a granularity-aware perspective. On the text side, unlike previous work that relies on single or ensemble text prompts with limited perception to align visual embeddings, we propose multi-perception

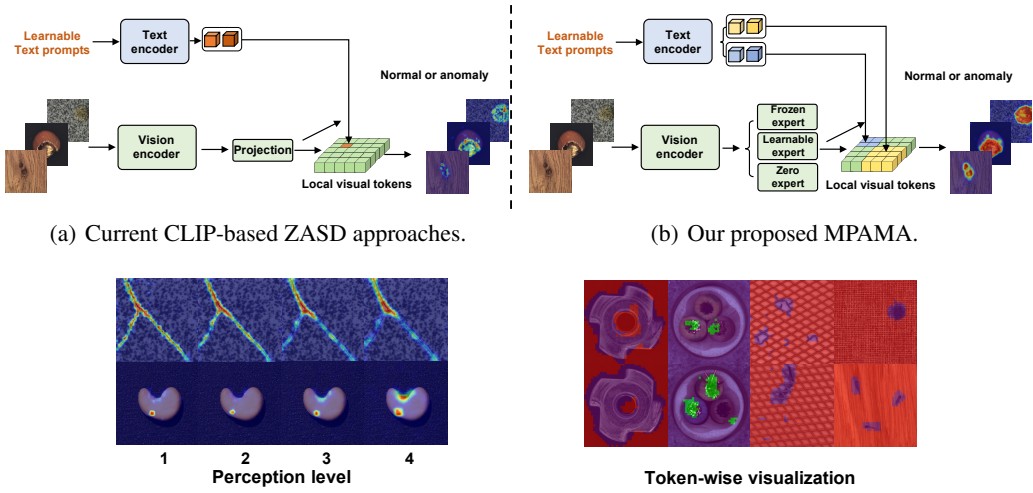

(a) Current CLIP-based ZASD approaches.    (b) Our proposed MPAMA.

(c) Segmentation visualization.    (d) Token-wise expert assignment.

Figure 1: Motivation of MPAMA. **(a)**: Current CLIP-based ZSAD methods typically rely on single-perception and coupled text prompts to align limited visual representations. **(b)**: In contrast, our approach introduces powerful textual and visual representations to mitigate the conflict between coarse-grained classification and fine-grained segmentation. **(c)**: Text prompts with small receptive fields (e.g., receptive fields of 1 and 2) are effective at detecting small-scale anomalies but are more susceptible to noise. In contrast, prompts with large perceptual fields are more robust to noise but tend to miss small anomalies. **(d)**: Experts capture distinct visual patterns. For example, the red expert primarily focuses on background and texture, the purple expert captures object regions and anomalies with large semantics variation, and the green expert tends to respond to deformations.

alignment that enriches text prompts with distinct receptive fields. This allows each prompt to be aware of structured and spatial anomalies at specialized scales. By ensembling these prompts, the model becomes capable of capturing anomalies across a wide range of scales in Figure 1(c).

Building on this, we aim to enhance visual granularity by replacing the traditional single shared expert with multiple specialized experts. To this end, we introduce Mixture-of-Experts adaptation (MA), which dynamically routes visual tokens to appropriate experts (Lepikhin et al., 2021). The selected experts, each with diverse specialized knowledge, jointly represent visual patch tokens at multiple levels of granularity, as illustrated in Figure 1(d). These components constitute the complete MPAMA framework. The main contributions of this paper are summarized as follows:

- We reveal that single-level alignment in existing ZSAD approaches hinders the model's ability to learn comprehensive anomaly semantics. To address this limitation, we propose MPA and MPAMA, which enhance visual-textual representation by extending coarse-grained alignment to multi-granularity alignment, enabling holistic anomaly learning.

- We expand traditional single-level alignment into a multi-level paradigm. Learnable text prompts are enriched with distinct perceptual fields to capture anomaly semantics at different receptive fields. These prompts are jointly optimized to incorporate anomaly semantics across multiple granularities.

- A single shared visual expert is insufficient to represent all visual patch tokens. To overcome this, we propose MA, which dynamically routes each patch token to multiple experts, each possessing specialized knowledge. The selected experts collaboratively represent the patch token from different granularities.

- Extensive experiments on datasets spanning a wide range of object categories have been conducted to demonstrate the effectiveness of MPA and MPAMA. The results consistently show superior performance in both anomaly detection and segmentation tasks.

## 2 PRELIMINARY

**Problem Formulation** Given an auxiliary dataset comprising normal and abnormal images $x \in \mathcal{R}^{3 \times H_{\text{image}} \times W_{\text{image}}}$ with their corresponding labels, i.e., $y \in \mathcal{R}$, $S \in \mathcal{R}^{H_{\text{image}} \times W_{\text{image}}}$, our objective is to train a zero-shot anomaly detection model capable of generalizing to previously unseen samples

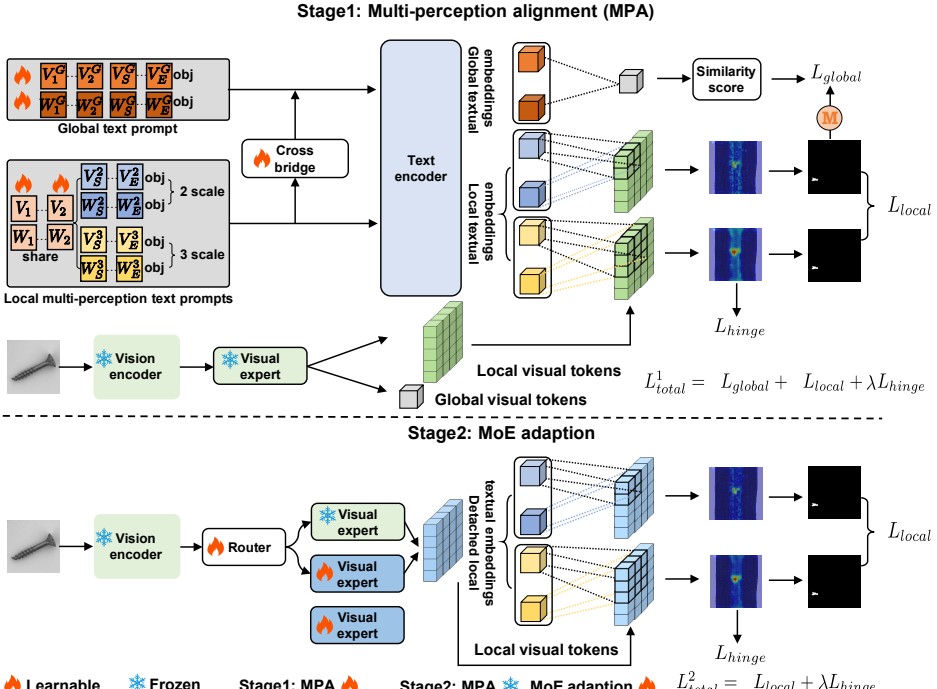

Figure 2: Overview of MPAMA. Our model consists of two stages. In Stage 1, we propose MPA to adapt the textual space of CLIP while keeping the visual space frozen. We introduce multi-perception text prompts to capture anomaly patterns at multiple spatial scales, enabling comprehensive semantic modeling. To further enhance representation, local and global text prompts are disentangled to mitigate the semantic coupling between fine-grained segmentation and coarse-grained classification. In Stage 2, we freeze the textual space and adapt the visual space by introducing specialized visual experts. These experts enhance the model's ability to represent complex and diverse anomaly patterns.

across different categories and domains. Note that 0 indicates normality and 1 indicates abnormality. The detection model is expected to perform anomaly classification $\mathcal{A}_c(x) : \mathcal{R}^{3 \times H_{\text{image}} \times W_{\text{image}}} \mapsto \mathcal{R}$ and anomaly segmentation, defined as $\mathcal{A}_s(x) : \mathcal{R}^{3 \times H_{\text{image}} \times W_{\text{image}}} \mapsto \mathcal{R}^{H_{\text{image}} \times W_{\text{image}}}$.

**A Brief Review of CLIP for Classification and Segmentation**   Given a learnable text prompt template $G$ for class $c$, the text encoder $T(\cdot)$ generates the corresponding textual embedding, denoted as $g_c = T(G_c) \in \mathcal{R}^D$. In the context of anomaly detection, the task involves classifying two classes: *normal* and *abnormal*. Accordingly, we define the text embeddings as $g_n = T(G_n)$ and $g_a = T(G_a)$, where $G_n$ and $G_a$ are the learnable prompts corresponding to the normal and abnormal classes, respectively. The vision encoder $F(\cdot)$ processes an input image $x$ to extract its visual representations $[\bar{f}, \bar{v}] = F(x)$. A projection layer $h(\cdot)$ projects them into class token $f = h(\bar{f})$ and patch tokens $v = h(\bar{v})$, where $f \in \mathcal{R}^D$ and $v \in \mathcal{R}^{H \times W \times D}$. These tokens are $\ell_2$-normalized by default. CLIP enables zero-shot anomaly detection by computing the similarity between textual and visual tokens. The image-level and patch-level abnormality possibilities are calculated as

$$P(g_a, f) = \frac{\exp(\langle g_a, f \rangle / \tau)}{\sum_{c \in \{n,a\}} \exp(\langle g_c, f \rangle / \tau)}, \qquad P(g_a, v^i) = \frac{\exp(\langle g_a, v^{(i,j)} \rangle / \tau)}{\sum_{c \in \{n,a\}} \exp(\langle g_c, v^i \rangle / \tau)},$$

where $\tau$ is a temperature hyperparameter and $\langle \cdot, \cdot \rangle$ denotes cosine similarity. We derive the segmentation of position $i$ as $S_a(i) = P(g_a, v_i)$, resulting in segmentation maps $S_n, S_a \in \mathcal{R}^{H \times W}$.

## 3  METHODS

**Overview**   This paper aims to enhance CLIP's visual-textual representations by extending coarse-grained alignment to multi-granularity alignment, enabling accurate zero-shot anomaly detection across diverse categories. As illustrated in Figure 2, the proposed framework consists of two main components: MPA and MA, which together adapt CLIP in both the textual and visual embedding

spaces. In the first stage (MPA), we adapt the textual space through MPA. In the second stage (MPAMA), we freeze the text encoder and adapt the visual space via MA.

## 3.1 Multi-perception alignment

Unseen anomalies typically appear at various scales and shapes. Prior works (Zhou et al., 2024a; Qu et al., 2025) that rely on single-level alignment often struggle to capture such diverse patterns. To address this limitation, we propose MPA, which captures anomalies of varying sizes and structures simultaneously. This enhances the model's generalization ability to detect a broader range of anomalies. Specifically, we define a set of learnable multi-scale text prompt templates $L = \{L^p\}_{p \in \mathcal{P}}$, where each $L^p = \{L_n^p, L_a^p\}$ contains the normality and abnormality prompts corresponding to perception level $p$, and $\mathcal{P}$ denotes the set of selected perception levels.

$$L_n^p = [V_1^p][V_2^p] \cdots [V_E^p][object] \qquad L_a^p = [W_1^p][W_2^p] \cdots [W_E^p][damaged][object].$$

where $[V_i^p]$ and $[W_i^p]$ ($V_i^p, W_i^p \in \mathcal{R}^D$) are learnable word embeddings in normality and abnormality text prompt templates of perception $p$, respectively. Each prompt is mapped through the text encoder to produce a scale-aware text embedding $\ell_c^p = T(L_c^p)$. To associate each text embedding with spatial visual context, we compute its similarity with an aggregated visual token at each spatial location $i$ using a generalized local aggregation operator. This provides each position with a broader receptive field, allowing the model to capture structural patterns, such as anomaly boundaries and large contiguous regions. In contrast, visual tokens with limited perceptual fields are prone to misdetecting large-scale anomalies due to insufficient contextual awareness. Formally, we define a generalized local aggregation operator $\mathcal{A}_p(v_i)$ over a $p \times p$ neighborhood $\mathcal{N}_p(i)$ centered at position $i$. The aggregated token $v_i^p$ is computed as:

$$v_i^p = \mathcal{AGG}_p(v_i) = \mathrm{AGG}\left(\{v_j \mid j \in \mathcal{N}_p(i)\}\right),$$

where $\mathcal{AGG}_p(\cdot)$ denotes a generic aggregation function used to summarize the local patch tokens. Common choices include average pooling, max pooling, or learnable alternatives such as attention mechanisms or small MLPs. In this work, we focus on average pooling and leave max pooling learnable aggregation strategies for future exploration. The segmentation at location $i$ under perception level $p$ is rewritten as:

$$S_a^p(i) = P(\ell_a, v_i^p) = \frac{\exp\left(\langle \ell_a^p, v_i^p \rangle / \tau\right)}{\sum_{c \in \{n,a\}} \exp\left(\langle \ell_c^p, v_i^p \rangle / \tau\right)}. \tag{1}$$

In doing so, $L^p$ is used to learn anomalies at scale $p$. We then ensemble text prompts with different perceptions to capture anomalies of varying sizes. To facilitate anomaly learning in different scales, we share the prefix portion of the learnable embeddings across text prompt templates with different perceptions. The resulting multi-perception prompt template is defined as:

$$L_n^p = [V_1] \cdots [V_S][V_{S+1}^p] \cdots [V_E^p][object], L_a^p = [W_1] \cdots [W_S][W_{S+1}^p] \cdots [W_E^p][damaged][object].$$

The $[V_i]$ and $[W_i]$ ($i \in 1, \ldots, S$) are shared learnable word embeddings in normality and abnormality text prompt templates across the perception levels.

We aim to simultaneously model global and local anomaly semantics for classification and segmentation. Existing works typically employ shared text prompts to learn global and local anomaly semantics. However, this coupling introduces an alignment compromise between the two objectives.

Anomaly classification requires prompt templates to identify an image as anomalous if it contains any abnormal region, even when most of the pixels are normal. In contrast, anomaly segmentation demands fine-grained prompts capable of distinguishing each pixel as normal or abnormal. From the perspective of classification, focusing too heavily on pixel-level semantics can impair the model's ability to capture high-level anomaly concepts, since anomalous images often contain large regions of normal content. Conversely, emphasizing global semantics can weaken the model's sensitivity to localized specific anomaly pixels, thereby impairing precise pixel-wise discrimination. This conflict indicates that using coupled prompt templates for both global and local tasks imposes a compromise, potentially degrading performance on both fronts. To resolve this, we propose to disentangle the prompts into two distinct branches: global prompts and local prompts, each optimized for its respective objective. Formally, we define the global text prompts as follows:

$$G_n = [V_1^G] \cdots [V_E^G][object], \ \ G_a = [W_1^G] \cdots [W_E^G][damaged][object]. \tag{2}$$

To further improve global semantic modeling, we introduce a cross-level bridging mechanism that transfers salient features of local prompts to the global prompt. This allows the global prompt to maintain a holistic view while benefiting from fine-grained local features.

Let $\bar{G}_a \in \mathcal{R}^{E \times D}$ and $\{\bar{L}_a^p\}_{p \in \mathcal{P}}$ be the learnable parts of global and local prompt embeddings for the abnormal class. We compute the element-wise maximum of $\{\bar{L}_a^p\}_{p \in \mathcal{P}}$ along the perception levels: $\bar{L}_a^{\max} = \max_p \{\bar{L}_a^p\}_{p \in \mathcal{P}}$. $\bar{L}_a^{\max} \in \mathcal{R}^{E \times D}$ is salient feature from local prompts. Then we integrate the salient feature with global prompt embeddings to the final $\tilde{G}_a$ by concatenating $\bar{G}_a$ with $\bar{L}_a^{\max}$ along the channel, followed by an MLP. The mathematical formulation is given by:

$$\tilde{G}_a = \text{MLP}\left(\text{Concat}(\bar{G}_a, \max_p\{\bar{L}_a^p\}_{p \in \mathcal{P}})\right), \quad \tilde{G}_n = \text{MLP}\left(\text{Concat}(\bar{G}_n, \max_p\{\bar{L}_n^p\}_{p \in \mathcal{P}})\right).$$

These lead to MPA, which adapts the textual space of CLIP for fine-grained anomaly modeling.

## 3.2 MULTI-GRANULARITY VISUAL REPRESENTATION VIA MoE ADAPTATION

Through MPA, the text space is adapted to handle anomalies with multiple levels of granularity. Here, we freeze the textual space and instead adapt the visual space to achieve multi-granular visual modeling. We treat the projection of the original visual encoder as the expert. Previous works replace the expert to learn a new expert from scratch. However, such an expert is limited in capturing the complex distribution and is easily overfitting to the auxiliary dataset, thus decreasing the generalization capacity. We propose MA to dynamically route visual tokens to multiple specialized experts, enabling joint representation of each patch token at multiple granularities. Unlike standard MoE approaches that rely on homogeneous experts, we define three types of experts: **learnable experts** $h^l$, **frozen experts** $h^f$, and **zero experts** $h^z(v_i) = 0$. For the frozen expert, the intuition is that certain pretrained experts already perform well on specific visual patterns and thus do not require further tuning. In contrast, learnable experts are essential for handling tokens with novel or unseen patterns, enabling the model to specialize in anomaly semantics through training. Lastly, some tokens could be sufficiently represented by a single expert. The zero expert provides a unified modeling mechanism by zeroing out the input.

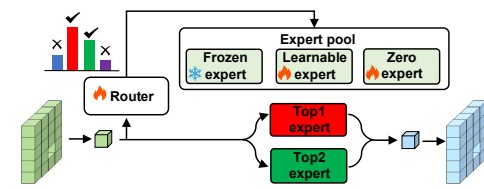

Figure 3: Overview of MOE adaptation.

Specifically, we first define the expert pool $\mathcal{H} = \{h_0^f, h_1^l, h_2^l, h_3^l, \cdots, h_N^z\}$. Considering the original expert contains generic prior knowledge, we use its weight as the initialization of all experts. Then, we use the trainable weight matrix $w \in \mathcal{R}^{D \times N}$ as the learnable router $\text{Ro}(\bar{v}_i) = w\bar{v}_i$, which decides to route tokens to specialized experts. $v$ is the final weighted sum of outputs from K activated experts:

$$v_i = \sum_{i=0}^{N} e_i h_i(\bar{v}_i), \quad e_i = \begin{cases} \text{Softmax}(\text{Ro}(\bar{v}_i)), & \text{if } \text{Ro}(\bar{v}_i) \in \text{Top-K}\left(\{\text{Ro}(\bar{v}_i) \mid 0 \le i \le N\}\right), \\ 0, & \text{otherwise.} \end{cases}$$

To sufficiently optimize on the router, we aggregate over $\bar{v}_i$ instead of $v_i$, i.e, $\bar{v}_i^p = \mathcal{AGG}_p(\bar{v}_i)$, and then feed them into the router, i.e., $\text{Ro}(\bar{v}_i^p) = w\mathcal{AGG}_p(\bar{v}_i), p \in \mathcal{P}$. The pre-aggregation increases the quantity and diversity of tokens input to the MoE compared to $\text{Ro}(\bar{v}_i)$ in the post-aggregation. The final output is rewritten as $v_i^p = \sum_{i=0}^{N} e_i h_i(\mathcal{AGG}_p(\bar{v}_i))$ different from $v_i^p = \mathcal{AGG}_p(\sum_{i=0}^{N} e_i h_i(\bar{v}_i))$ in the post-aggregation.

**Multiple-perception Context Optimization** Given the global text prompts $\tilde{G}_a$ and $\tilde{G}_n$, we could obtain the text embeddings $\tilde{g}_a$ and $\tilde{g}_n$ for abnormality and normality. We define $L_{global}$ as a cross-entropy loss to align the image-level anomalies:

$$L_{global} = \text{CrossEntroy}([P(\tilde{g}_n, f), P(\tilde{g}_a, f)], y) \tag{3}$$

Given the local text prompts $\{L_a^p\}_{p \in \mathcal{P}}$ and $\{L_n^p\}_{p \in \mathcal{P}}$, our text prompts learn pixel-level multi-perception anomaly semantics by minimizing the following local loss:

$$L_{local} = \sum_{p \in \mathcal{P}} \text{Focal}(\text{Up}([S_n^p, S_a^p]), S) + \text{Dice}(\text{Up}(S_n^p), I - S) + \text{Dice}(\text{Up}(S_a^p), S), \tag{4}$$

where $\text{Focal}(\cdot)$, $\text{Dice}(\cdot)$, and $\text{Up}(\cdot)$ denote the Focal loss (Lin et al., 2017), Dice loss (Li et al., 2019), and upsampling interpolation operation, respectively. To better separate normal and abnormal regions, we employ a hinge loss that encourages a larger margin between their respective anomaly scores:

$$L_{\text{hinge}} = \sum_{p \in \mathcal{P}} \left(\frac{1}{|\mathcal{N}_0|} \sum_{i \in \mathcal{N}_0} \max\left(S_a^p(i) - \delta_n, 0\right) + \frac{1}{|\mathcal{N}_1|} \sum_{i \in \mathcal{N}_1} \max\left(\delta_a - S_a^p(i), 0\right)\right),$$

where $\mathcal{N}_0 = \{(i) \mid S(i) = 0\}$ and $\mathcal{N}_1 = \{(i) \mid S(i) = 1\}$ denote the sets of normal and anomalous pixels, respectively. $\delta_n \in \mathcal{R}$ and $\delta_a \in \mathcal{R}$ are the margin thresholds for normal and anomaly regions. This loss ensures that scores in normal regions remain below $\delta_n$, while encouraging anomaly scores in anomalous regions to exceed $\delta_a$.

MPAMA consists of two main components that adapt CLIP in both textual and visual spaces: MPA and MA. We adopt a two-stage training framework. Stage 1: Adapt the textual space via Multi-perception prompt learning, resulting in MPA. The total loss $L_{\text{total}}$ jointly optimizes both local and global semantics with hinge loss regularization:

$$L_{total}^1 = L_{global} + L_{local} + \lambda L_{hinge}. \tag{5}$$

Stage 2: The textual space is frozen, and the visual space is adapted using MA to align local semantics. This leads to the complete MPAMA. The corresponding loss is defined as:

$$L_{total}^2 = L_{local} + \lambda L_{hinge}, \tag{6}$$

where $\lambda$ is a weighting coefficient used to balance the three loss components.

**Inference**   During Inference, we compute the final anomaly segmentation by averaging the segmentation maps across all perception levels: $\mathcal{A}_s(x) = \frac{1}{|\mathcal{P}|} \sum_{p \in P} S_a^p$. As for image-level anomaly detection, we integrate the global text similarity score and the maximum of the segmentation map: $\mathcal{A}_c(\cdot) = \frac{1}{2} P(\tilde{g_a}, f) + \frac{1}{2} \max(S_a)$.

## 4 EXPERIMENTS

### 4.1 EXPERIMENT SETUP

**Dataset Details & Baselines**   To comprehensively demonstrate the generalized zero-shot anomaly detection capability of MPAMA, we conducted extensive evaluations on publicly available datasets spanning a wide range of semantic categories across industrial and medical domains. For industrial inspection, we include MVTec AD (Bergmann et al., 2019), VisA (Zou et al., 2022), MPDD (Jezek et al., 2021), BTAD (Mishra et al., 2021), SDD (Tabernik et al., 2020), DAGM (Wieler & Hahn, 2007), and DTD-Synthetic (Aota et al., 2023). In the medical domain, we evaluate on datasets for skin cancer diagnosis (ISIC (Gutman et al., 2016)), colon polyp detection (CVC-ClinicDB (Bernal et al., 2015), CVC-ColonDB (Tajbakhsh et al., 2015), Kvasir (Jha et al., 2020), Endo (Hicks et al., 2021)), brain tumor detection (HeadCT (Salehi et al., 2021), BrainMRI (Salehi et al., 2021), and Br35H (Hamada., 2020)). Dataset and baseline details see Appendix A and B. $^\dagger$ denotes results taken from original papers.

**Evaluation Setting and Metrics**   We report AUROC and Average Precision (AP) to measure detection performance, and use AUROC and AUPRO to evaluate segmentation performance. All metrics are averaged over five runs. Following AnomalyCLIP, we use the MVTec AD dataset as the auxiliary training set for evaluating other datasets, and vice versa, training on VisA for MVTec AD.

**Implementation Details**   We use the publicly available CLIP model (`ViT-L/14@336px`) as the backbone. We follow the data processing of AnomalyCLIP. All images are resized to $518 \times 518$. The length of the learnable embedding is set to $E = 12$, with a shared prefix length of $S = 2$. The perception set is defined as $\mathcal{P} = \{1, 2, 3, 4\}$. We use the top feature of the vision encoder as the visual representations, and use average pooling as the aggregation operator $\mathcal{AGG}$. The number of experts is set to 4 for training on VisA and 5 for MVTec AD, with 2 activated experts. We set the hinge loss thresholds to $\delta_a = \delta_n = 0.5$, and the weighting coefficients to $\lambda = 5$. We also provide the ablation study of $\lambda$ in Appendix F. We use Adama optimizer with a learning rate of 1e-3 in stage 1 and 1e-6 in stage 2. The batch size is set to 8, and the training epoch is 30 in stage 1 and 10 in stage 2. All experiments are conducted using PyTorch 2.0.0 on a single NVIDIA RTX 3090 (24GB) GPU.

### 4.2 MAIN RESULTS

**Generalized ZSAD for Industrial Anomaly Detection**   We compare our models, MPA and MPAMA, with SOTA ZSAD baselines, including WinCLIP, VAND, CoOp, AdaCLIP, and Anomaly-CLIP, across seven industrial datasets covering over 60 semantic categories. As shown in Table 1, our models consistently outperform baselines in both anomaly detection and segmentation tasks. For image-level detection, on the MVTec AD dataset, MPA achieves 94.1% AUROC and 96.9% AP, surpassing AnomalyCLIP's 91.5% AUROC and 96.2% AP. On the VisA dataset, MPA reaches

Table 1: ZSAD performance on industrial domain datasets. Best: Red; Second-best: Blue. † denotes results taken from original papers.

| Task | Dataset | WinCLIP (CVPR'23) | VAND (ARXIV'23) | AdaCLIP (ECCV'24) | AnomalyCLIP (ICLR'24) | FAPromt (ICCV'25) | MPA (Ours) | MPAMA (Ours) |
|---|---|---|---|---|---|---|---|---|
| Image-level (AUROC, AP) | MVTec AD | (91.8, 96.5)† | (86.1, 93.5)† | (89.6, - )† | (91.5, 96.2)† | (90.8, 94.9) | (94.1, 96.9) | (94.5, 97.6) |
| | VisA | (78.1, 81.2)† | (78.0, 81.4)† | (83.8, - )† | (82.1, 85.4)† | (83.6, 85.6) | (85.0, 87.6) | (85.4, 88.0) |
| | MPDD | (63.6, 69.9) | (73.0, 80.2) | (76.8, - )† | (77.0, 82.0)† | (77.5, 82.2) | (79.4, 81.4) | (81.2, 82.1) |
| | BTAD | (68.2, 70.9) | (73.6, 68.6) | (88.6, - )† | (88.3, 87.3)† | (92.3, 93.0) | (89.0, 88.2) | (89.8, 88.9) |
| | SDD | (84.3, 77.4) | (79.8, 71.4) | ( - , - )† | (84.7, 80.0)† | (81.8, 77.5) | (84.9, 80.8) | (85.2, 81.2) |
| | DAGM | (91.8, 79.5) | (94.4, 83.8) | (98.3, - )† | (97.5, 92.3)† | (96.9, 90.6) | (98.4, 92.9) | (98.5, 93.2) |
| | DTD-Synthetic | (93.2, 92.6) | (86.4, 95.0) | (95.5, - )† | (93.5, 97.0)† | (95.6, 97.4) | (96.2, 98.1) | (97.0, 98.8) |
| Pixel-level (AUROC, PRO) | MVTec AD | (85.1, 64.6)† | (87.6, 44.0)† | (90.3, - )† | (91.1, 81.4)† | (90.6, 81.6) | (92.5, 87.5) | (93.2, 88.3) |
| | VisA | (79.6, 56.8)† | (94.2, 86.8)† | (95.6, - )† | (95.5, 87.0)† | (95.6, 86.7) | (95.8, 87.9) | (96.1, 88.6) |
| | MPDD | (76.4, 48.9) | (94.1, 83.2) | (96.4, - )† | (96.5, 88.7)† | (95.7, 85.6) | (96.5, 88.3) | (96.8, 89.2) |
| | BTAD | (72.7, 27.3) | (60.8, 25.0) | (92.1, - )† | (94.2, 74.8)† | (94.8, 75.1) | (95.3, 78.5) | (96.0, 79.1) |
| | SDD | (68.8, 24.2) | (79.8, 65.1) | ( - , - )† | (90.6, 67.8)† | (92.5, 70.0) | (92.7, 73.9) | (93.9, 74.2) |
| | DAGM | (87.6, 65.7) | (82.4, 66.2) | (91.0, - )† | (95.6, 91.0)† | (98.1, 94.9) | (96.6, 91.9) | (97.0, 92.7) |
| | DTD-Synthetic | (83.9, 57.8) | (95.3, 86.9) | (96.9, - )† | (97.9, 92.3)† | (98.0, 92.2) | (98.3, 92.9) | (98.5, 93.8) |

85.0% AUROC and 87.6% AP, outperforming both 83.8% AUROC of AdaCLIP and 82.1% AUROC and 85.4% AP of AnomalyCLIP. We attribute this superiority to fine-grained anomaly semantics modeling at multiple scales and disentanglement learning of global and local anomaly semantics. Building on MPA, MPAMA further improves performance, reaching 94.5% AUROC and 97.6% AP on MVTec AD, and 85.4% AUROC and 88.0% AP on VisA. These gains stem from multiple specialized visual experts through MoE adaptation, which enhances the model's capacity to represent complex and diverse anomaly patterns. In terms of anomaly segmentation, our model achieves consistent superiority. On MVTec AD, MPA achieves 92.5% AUROC and 87.5% PRO, while MPAMA improves this to 93.2% AUROC and 88.3% PRO, outperforming the previous best by 91.1% AUROC and 81.4% PRO of AnomalyCLIP. Similar trends are observed on MPDD, BTAD, and so on. These illustrated that our models could learn the generalized anomaly semantics from multi-scale anomalies.

Table 2: Cross-domain ZSAD performance on medical analysis. Best: Red; Second-best: Blue.

| Task | Dataset | WinCLIP (CVPR'23) | VAND (ARXIV'23) | AdaCLIP (ECCV'24) | AnomalyCLIP (ICLR'24) | FAPromt (ICCV'25) | MPA (Ours) | MPAMA (Ours) |
|---|---|---|---|---|---|---|---|---|
| Image-level (AUROC, AP) | HeadCT | (81.8, 80.2) | (89.1, 89.4) | (91.5, - ) | (93.4, 91.6) | (93.9, 92.6) | (96.2, 96.0) | (96.4, 96.2) |
| | BrainMRI | (86.6, 91.5) | (89.3, 90.9) | (94.8, - ) | (90.3, 92.2) | (94.8, 93.7) | (95.0, 95.8) | (95.6, 96.0) |
| | Br35H | (80.5, 82.2) | (93.1, 92.9) | (97.7, - ) | (94.6, 94.7) | (96.6, 95.6) | (97.9, 97.9) | (98.1, 98.0) |
| Pixel-level (AUROC, PRO) | ISIC | (83.3, 55.1) | (89.4, 77.2) | (88.3, - ) | (89.7, 78.4) | (90.7, 80.3) | (92.7, 85.2) | (93.5, 85.8) |
| | ColonDB | (70.3, 32.5) | (78.4, 64.6) | (79.1, - ) | (81.9, 71.3) | (84.1, 73.2) | (83.9, 72.4) | (84.9, 72.6) |
| | ClinicDB | (51.2, 13.8) | (80.5, 60.7) | (84.4, - ) | (82.9, 67.8) | (83.9, 69.3) | (84.6, 70.5) | (84.9, 70.3) |
| | Kvasir | (69.7, 24.5) | (75.0, 36.2) | ( - , - ) | (78.9, 45.6) | (80.4, 46.5) | (80.3, 45.7) | (81.9, 46.6) |
| | Endo | (68.2, 28.3) | (81.9, 54.9) | ( - , - ) | (84.1, 63.6) | (85.9, 65.0) | (87.1, 68.8) | (87.7, 69.5) |

**Cross-Domain Generalization from Industrial to Medical Domains**   The above results demonstrate that our models generalize well to industrial datasets. To further assess the generalization capability, we explore a more challenging cross-domain setting: train our model using the MVTec AD industrial dataset and directly test it on various medical tasks. As shown in Table 2, our models exhibit strong cross-domain generalization. Notably, for brain tumor detection, MPA achieves 96.2% AUROC on BrainMRI, while MPAMA further improves performance on HeadCT, surpassing AnomalyCLIP by increasing AUROC from 93.4% to 96.4% and AP from 91.6% to 96.2%. In the case of skin cancer detection, our models also demonstrate significant improvement. These results highlight the effectiveness and robustness of our proposed MPA and MPAMA frameworks, which generalize well beyond the source domain and adapt effectively to unseen domains.

### 4.3   RESULT ANALYSIS

**Multi-perception Prompt Learning vs. Single-perception Prompting Learning**   Here, we aim to investigate whether text prompts with different receptive fields can effectively capture anomalies at varying scales. We conduct a quantitative analysis to demonstrate our model's superior ability to detect anomalies across different scales. Specifically, we first count the number of anomaly pixels in each image from the MVTec AD dataset and then divide the images into four scale categories—*tiny*, *small*, *medium*, and *large*—based on quartile thresholds. As shown in Figure 4, MPAMA achieves performance comparable to AnomalyCLIP for tiny and small-scale anomalies, illustrating that our method does not compromise performance on small anomalies. For medium and large anomalies,

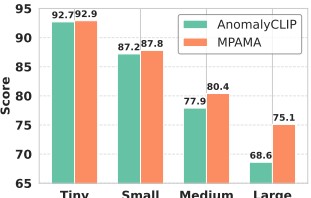
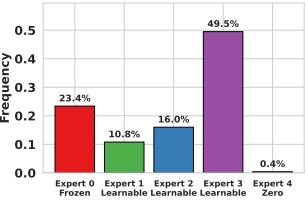

Figure 4: Scale-wise results between AnomalyCLIP and MPAMA on MVTec. **Left**: Pixel-level AUROC; **Right**: AUPRO.

Figure 5: Frequency of expert usage on MVTec AD.

MPAMA outperforms AnomalyCLIP by a significant margin, with the performance gain increasing for larger anomalies. In summary, integrating multi-scale receptive field prompts provides an effective strategy for balancing detection across different anomaly scales while enhancing robustness to noise. More visualizations are provided in Appendix 7.

**Selection frequency of Multiple Specialized Experts**   MPAMA leverages multiple specialized visual experts to comprehensively model the visual distribution. We analyze the frequency of expert usage across the MVTec AD dataset, as shown in Figure 5. Since some visual tokens can be well represented by the original expert, approximately 20% of tokens are routed to the frozen expert. Nearly 80% of tokens are routed to the learnable experts for adaptation, highlighting the necessity of fine-tuning beyond the frozen representation. In particular, we observe that the purple expert accounts for 50% of the routing in Figure 1(d), primarily capturing object semantics and large-scale variations. The blue expert receives 16% of the tokens, while the green expert accounts for approximately 10%. Additionally, 0.4% of tokens are routed to zero experts, indicating that very few tokens can be adequately represented by a single expert alone. It also reflects the importance of projecting tokens using multiple experts. Although all of these are learnable experts, each specializes in capturing distinct visual patterns. These demonstrate that MPAMA adaptively routes tokens to different experts based on their semantic characteristics, enabling more specialized and expressive feature modeling.

# 5   ABLATION STUDY

**Module Ablation**   We first investigate the contribution of three key modules to the overall performance of MPAMA: multi-perception text prompts ($T_1$), decoupled local and global text prompts ($T_2$), and MA ($T_3$). Table 3 shows that incorporating multi-perception text prompts significantly improves both image-level anomaly detection and pixel-level segmentation by capturing multi-scale anomaly

Table 3: Ablation on proposed modules.

| $T_1$ | $T_2$ | $T_3$ | MVTec AD Pixel-level | MVTec AD Image-level | VisA Pixel-level | VisA Image-level |
|---|---|---|---|---|---|---|
| | | | (91.1, 81.4) | (91.5, 96.2) | (95.5, 87.0) | (82.1, 85.4) |
| ✓ | | | (92.1, 86.4) | (93.0, 96.9) | (95.6, 87.5) | (83.2, 87.1) |
| ✓ | ✓ | | (92.5, 87.5) | (94.1, 96.9) | (95.8, 87.9) | (85.0, 87.6) |
| ✓ | | ✓ | (92.3, 86.0) | (93.1, 96.5) | (95.8, 88.1) | (84.6, 87.3) |
| | ✓ | ✓ | (91.4, 83.8) | (92.2, 96.3) | (95.6, 87.4) | (83.5, 86.7) |
| ✓ | ✓ | ✓ | (93.2, 88.3) | (94.5, 97.6) | (96.1, 88.6) | (85.4, 88.0) |

semantics. When we further decouple the global and local text prompts, an additional performance gain is observed, highlighting the benefits of reducing semantic interference between coarse and fine-grained representations. Finally, after adapting the textual space, we introduce MA to dynamically route visual tokens to specialized experts. This allows MPAMA to comprehensively learn complex anomaly patterns compared to MPA.

**Ablation on Perception Number**   In this section, we investigate the impact of varying the perception number. As shown in Table 4, increasing the number of perception levels consistently improves both image-level and pixel-level performance. The model achieves optimal performance when the perception set is 1, 2, 3, 4. However, further

Table 4: Perception number ablation.

| Perception number | MVTec AD Pixel-level | MVTec AD Image-level | VisA Pixel-level | VisA Image-level |
|---|---|---|---|---|
| {1} | (92.1, 87.2) | (93.0, 96.8) | (95.7, 87.9) | (84.6, 86.9) |
| {1,2} | (92.3, 87.1) | (93.9, 97.3) | (95.7, 88.0) | (84.6, 87.1) |
| {1,2,3} | (92.3, 87.7) | (94.0, 97.5) | (95.8, 88.3) | (85.1, 87.6) |
| {1,2,3,4} | (93.2, 88.3) | (94.5, 97.6) | (96.1, 88.6) | (85.4, 88.0) |
| {1,2,3,4,5} | (92.7, 87.7) | (94.4, 97.7) | (95.8, 88.2) | (85.6, 87.9) |

increasing the perception number to include a receptive field of 5 leads to a performance drop. This is due to overly large receptive fields interfering with local information modeling. Therefore, an appropriate number of perception levels is crucial for effectively capturing diverse anomaly semantics.

**Ablation on Expert Number**   The number of experts determines model's capacity to learn specialized visual representations. As shown in Table 5, increasing the number of experts from 1 to 3 consistently improves performance on both anomaly detection and segmentation tasks. This indicates that utilizing multiple experts enables the model to capture more complex and diverse anomaly patterns.

Table 5: Expert number ablation.

| Expert number | MVTec AD Pixel-level | MVTec AD Image-level | VisA Pixel-level | VisA Image-level |
|---|---|---|---|---|
| 1 | (92.1, 86.4) | (93.0, 96.9) | (95.6, 87.5) | (83.2, 87.1) |
| 2 | (92.3, 87.0) | (93.0, 97.1) | (95.8, 87.8) | (84.1, 87.5) |
| 3 | (92.8, 87.8) | (94.1, 97.3) | (96.0, 88.1) | (85.0, 87.8) |
| 4 | (93.2, 88.3) | (94.5, 97.6) | (96.1, 88.4) | (85.1, 88.0) |
| 5 | (93.0, 87.9) | (94.6, 97.8) | (96.1, 88.6) | (85.4, 88.0) |
| 6 | (92.9, 87.9) | (94.7, 97.8) | (96.2, 88.3) | (85.3, 87.9) |

However, when the number of experts increases to 6, the performance improvement becomes marginal compared to using 4 or 5 experts. Hence, an appropriate number of experts improves the learning of anomaly patterns during model training on the auxiliary dataset.

## 6 RELATED WORK

**Zero-shot Anomaly Detection** Recently, zero-shot anomaly detection (ZSAD) has emerged as an active research area, attracting increasing attention from AD community (Aota et al., 2023; Liznerski et al., 2022; Esmaeilpour et al., 2022; Zhou et al., 2024a; Chen et al., 2023; Zhou et al., 2024b). A promising research line is to leverage CLIP as the backbone for ZSAD (Jeong et al., 2023; Chen et al., 2023; Zhou et al., 2024a). WinCLIP designs human-crafted normal and abnormal text prompts for zero-shot anomaly detection without fine-tuning. AnomalyCLIP (Zhou et al., 2024a) proposes object-agnostic prompt learning and demonstrates strong potential across diverse domains. Subsequent methods enhance performance via ensemble prompt strategies for modeling more anomaly semantics such as FAprompt (ZHU et al., 2025) and BayesCLIP (Qu et al., 2025), or integrate visual adaptation techniques to refine the alignment between visual and textual features such as VCP-CLIP (Qu et al., 2024), AdaptCLIP (Cao et al., 2024), and AA-CLIP (Ma et al., 2025). However, most methods rely on single-scale text embeddings and coupled alignment strategies. This coarse alignment hinders the model's ability to capture the inherently multi-scale nature of anomalies. Our approach introduces multi-perception prompt learning, which assigns varied receptive fields to text prompts. Combined with a fine-grained expert assignment mechanism, this enables precise anomaly detection across multiple spatial scales. MLLM-based methods (Li et al., 2023; Gu et al., 2024b; Xu et al., 2025) demonstrate that fine-tuned MLLMs can perform basic reasoning on detected anomalies. However, these approaches require significantly larger backbones and substantially more training data.

**Prompt Learning in Anomaly Detection** Prompt learning aims to achieve lightweight domain adaptation with minimal training overhead by updating only a small set of trainable parameters. CoOp (Zhou et al., 2022b;a) first introduced this approach in the context of few-shot classification by learning task-specific prompts for CLIP. AnomalyCLIP (Zhou et al., 2024a) was the first to extend prompt learning to anomaly detection. It introduces object-agnostic prompt tuning to discard explicit class semantics, promoting more generalized anomaly representations. PromptAD (Li et al., 2024), FILo (Gu et al., 2024a), and BayesCLIP (Qu et al., 2025) propose hybrid prompts that combine learnable text embeddings with prior textual knowledge. However, existing methods typically employ a single-scale text prompt to model both fine-grained local anomalies and high-level global semantics. In contrast, we identify that such a coupled single-scale prompt is insufficient for capturing the diverse nature of anomaly semantics. To address this, we propose multi-perception text prompts, which assign diverse receptive fields to different prompts. Furthermore, we disentangle local and global anomaly modeling and enable more comprehensive anomaly semantics.

## 7 CONCLUSION

In this paper, we identify two key shortcomings in existing CLIP-based approaches stemming from single-level alignment. To address these challenges, we enhance visual-textual representations by extending single-level alignment to multi-level alignment. Specifically, we propose a multi-granularity alignment framework that strengthens cross-modal alignment: Multi-perception alignment enriches the textual space by attaching prompts with diverse receptive fields; MoE adaptation dynamically routes visual tokens to multiple specialized experts, enabling multi-granularity, token-wise representation. Based on this framework, we develop two models, MPA and MPAMA, capable of capturing rich and multi-granularity anomaly semantics. Extensive experiments on standard benchmarks demonstrate the superior performance of our approaches.

**Limitations** While MPAMA achieves strong performance through multiple specialized experts, this design can lead to increased computational overhead as the number of experts grows. However, the additional cost remains low in practice, since only two experts are activated per token during inference. A detailed analysis of memory consumption is provided in Appendix C.

**Broader Impacts** This paper proposes a new perspective on fine-tuning vision-language models that is fully compliant with legal and regulatory guidelines. While current prompt-learning methods typically rely on single-level alignment, our approach introduces a multi-level alignment strategy capable of capturing structural semantics more effectively. We hope this work draws greater attention to the role of receptive fields in the adaptation of vision-language models.

## REPRODUCIBILITY STATEMENT

We provide dataset details in Appendix A and baseline descriptions in Appendix B. Further analyses are included in the following appendices: computational overhead in Appendix C, prompt design strategies in Appendix E, weight coefficient sensitivity in Appendix F, and perception level ablation in Appendix G. To offer a deeper understanding of MPAMA's behavior, failure cases are presented in Appendix H. Additionally, subset-level quantitative results and visualizations are provided in Appendix I. **The code will be made available once accepted.**

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

## A  DATASETS

In this section, we present the statistical summary of the datasets used in our study. The 15 datasets span both industrial and medical domains. The details are reported in Table A.

Table 6: Overview of datasets used for anomaly and disease detection across industrial and medical domains.

| Dataset | Domain | Modality | $|\mathcal{C}|$ | Normal / Anomalous | Application |
|---------|--------|----------|-----|--------------------|-------------|
| *Industrial Inspection* | | | | | |
| MVTec AD | Object & Texture | RGB | 15 | (467, 1258) | Defect detection |
| VisA | Object | RGB | 12 | (962, 1200) | Defect detection |
| MPDD | Object | RGB | 6 | (176, 282) | Defect detection |
| BTAD | Object | RGB | 3 | (451, 290) | Defect detection |
| SDD | Object | RGB | 1 | (181, 74) | Defect detection |
| DAGM | Texture | RGB | 10 | (6996, 1054) | Defect detection |
| DTD-Synthetic | Texture | RGB | 12 | (357, 947) | Defect detection |
| *Skin Lesion Analysis* | | | | | |
| ISIC | Skin | RGB | 1 | (0, 379) | Skin cancer detection |
| *Colon Polyp Detection* | | | | | |
| ClinicDB | Colon | Endoscopy | 1 | (0, 612) | Polyp detection |
| ColonDB | Colon | Endoscopy | 1 | (0, 380) | Polyp detection |
| Kvasir | Colon | Endoscopy | 1 | (0, 1000) | Polyp detection |
| Endo | Colon | Endoscopy | 1 | (0, 200) | Polyp detection |
| *Thyroid Nodule Detection* | | | | | |
| *Brain Tumor Detection* | | | | | |
| HeadCT | Brain | CT | 1 | (100, 100) | Tumor detection |
| BrainMRI | Brain | MRI | 1 | (98, 155) | Tumor detection |
| Br35H | Brain | MRI | 1 | (1500, 1500) | Tumor detection |

## B  BASELINES

To comprehensively evaluate the performance of our model, we benchmark it against a range of ZSAD approaches. **We exclude the approaches based on large multimodal models (Gu et al., 2024b; Xu et al., 2025; Li et al., 2023) from comparison due to their use of stronger backbones and substantially more data**. We compare our method with SOTA ZSAD approaches, WinCLIP (Jeong et al., 2023), VAND (Chen et al., 2023), AnomalyCLIP (Zhou et al., 2024a), AdaCLIP (Cao et al.,

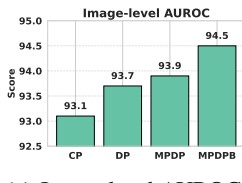 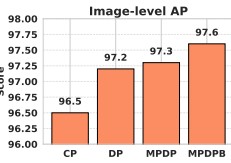 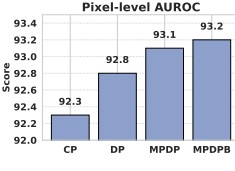 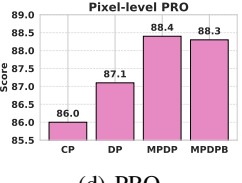

(a) Image-level AUROC.  (b) Image-level AP.  (c) Pixel-level AUROC.  (d) PRO.

Figure 6: Comparison using different text prompts. We denote coupling prompt, decoupling prompt, multi-perception prompt, and multi-perception prompt with cross bridge as CP, DP, MPDP, and MPDPB, respectively.

2024), and FAprompt (ZHU et al., 2025). Below is an overview of the compared approaches and the strategies used for adaptation:

- **WinCLIP (CVPR 2023)** (Jeong et al., 2023): A leading framework for zero-shot anomaly segmentation that introduces a rich set of prompts crafted for anomaly scenarios. It further integrates a window-based scaling mechanism to improve localization precision. All experimental settings replicate those reported in the original paper.

- **VAND (ARXIV 2023)** (Chen et al., 2023): VAND refines the prompt design and introduces learnable linear transformations to better capture fine-grained visual details. We adopt the original configuration to ensure comparability with reported results.

- **AdaCLIP (ECCV 2024)** (Cao et al., 2024): AdaCLIP uses the human-crafted text prompts as the text embedding, and instead adapts the visual space for ZSAD.

- **AnomalyCLP (ICLR 2024)** (Zhou et al., 2022b): AnomalyCLIP introduces an object-agnostic prompt learning framework to enhance generalization across diverse anomaly types. In addition, it integrates a DPAM to refine visual feature space of CLIP for more precise detection and segmentation

- **FAPromt (ICCV 2025)** (ZHU et al., 2025): FAPrompt ensembles multiple learnable text prompts to learn the comprehensive anomaly semantics.

## C  COMPUTATION OVERHEAD

Table 7: Analysis of computation overhead.

| Methods | Inference Time (s) | FPS | Peak GPU Memory (MB) | MVTec AD | | VisA | |
|---|---|---|---|---|---|---|---|
| | | | | pixel-level | image-level | pixel-level | image-level |
| AnomalyCLIP | 0.124 | 8.04 | 2235MB | (91.1, 81,4) | (91.5, 96.2) | (95.5, 87.0) | (82.1, 85.4) |
| MPA | 0.137 | 7.29 | 3275MB | (92.5, 87.5) | (94.1, 96.9) | (95.8, 87.9) | (85.0, 87.6) |
| MPAMA | 0.152 | 6.56 | 3275MB | (93.2, 88.3) | (94.5, 97.6) | (96.1, 88.6) | (85.4, 88.0) |

Besides accuracy, computational overhead is also a crucial metric for evaluating the performance of anomaly detection algorithms. In this section, we investigate whether MPAMA can effectively balance detection performance and computational cost. To this end, we measure GPU memory consumption during training and inference speed in terms of FPS. All experiments are conducted on an idle NVIDIA A100 GPU with a batch size of 1. The results are summarized in Table 7. Compared to AnomalyCLIP, both MPA and MPAMA incur slightly higher inference time and GPU memory usage. Specifically, the inference time of AnomalyCLIP and MPAMA is 0.124s and 0.152s, respectively. Despite this negligible increase in inference time, our method achieves notable improvements in anomaly detection and segmentation performance.

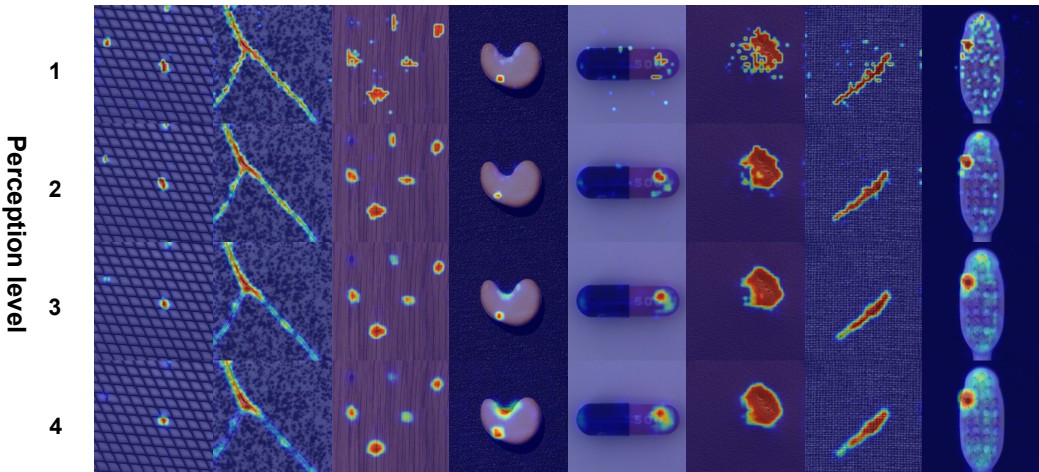

Figure 7: Visualization results using text prompts with varying receptive fields. From top to bottom, the receptive field size increases from 1 to 4.

## D    VISUALIZATION WITH DIFFERENT PERCEPTION LEVELS.

## E    ANALYSIS OF DIFFERENT PROMPT STRATEGIES.

Here, we analyze the effect of using different types of text prompts. We explore four configurations: Coupling Prompt (CP), which uses a single text prompt to jointly model both local and global semantics; Decoupling Prompt (DP), which uses two independent prompts to separately model global and local semantics; Multi-Perception Decoupling Prompt (MPDP), which extends DP by assigning a unique, unshared prompt to each perception scale; and Multi-Perception Decoupling Prompt with Bridge (MPDPB), which further introduces a cross-bridge mechanism to incorporate salient local semantic features into the global context. All variants are optimized using the same loss function described in Equ. 5 and Equ. 6. As shown in Figure 6, DP provides a consistent improvement across both image-level and pixel-level metrics, indicating that fully coupling global and local prompts hinders accurate anomaly modeling. MPDP takes this further by decoupling the local prompt across multiple perception scales. This further decoupling benefits pixel-level detection. Finally, MPDPB enhances global semantic modeling by integrating salient local features through the cross-bridge, leading to gains in image-level metrics.

## F    HYPERPARAMETER ABLATION

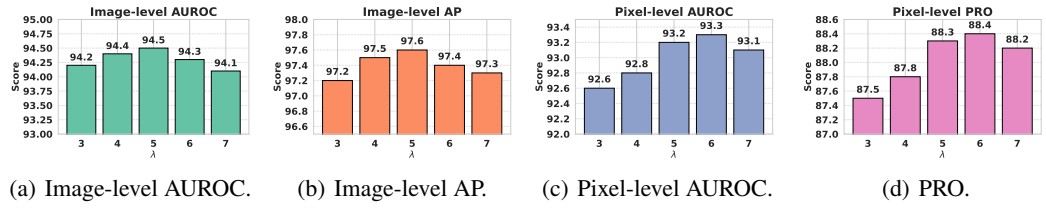

(a) Image-level AUROC.          (b) Image-level AP.          (c) Pixel-level AUROC.          (d) PRO.

Figure 8: The effect of $\lambda$

Here, we investigate the effect of the coefficient $\lambda$ on the model's performance. We set $\lambda \in \{3, 4, 5, 6, 7\}$. As shown in Figure 8, increasing $\lambda$ initially improves both image-level and pixel-level performance. However, as $\lambda$ continues to increase, performance begins to decline. Empirically, $\lambda = 5$ achieves a good balance between image-level and pixel-level performance.

Table 8: Ablation study on perception level.

| Perception | MVTec AD | | VisA | |
| level | Pixel-level | Image-level | Pixel-level | Image-level |
|---|---|---|---|---|
| {1,2,3,4} | (93.2, 88.3) | (94.5, 97.6) | (96.1, 88.6) | (85.4, 88.0) |
| {2,3,4,5} | (92.8, 87.5) | (94.4, 97.7) | (95.9, 88.0) | (85.1, 87.6) |
| {3,4,5,6} | (92.3, 86.1) | (94.3, 97.1) | (95.7, 87.5) | (84.6, 86.8) |

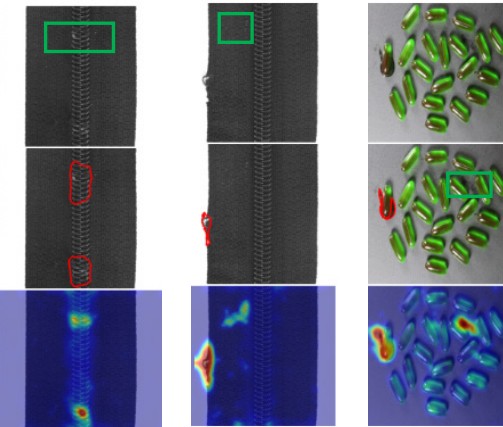

Figure 9: Illustration of false detection cases. Misdetected regions are marked with green squares.

## G  ABLATION ON PERCEPTION LEVEL

We further explore the influence of the perception scale while keeping the number of perception levels fixed. As shown in Table 8, the configuration $\{1, 2, 3, 4\}$ achieves the best performance compared to alternatives such as $\{2, 3, 4, 5\}$ and $\{3, 4, 5, 6\}$. These results highlight that small-scale perception (i.e., receptive field 1) is crucial for fine-grained anomaly detection. In contrast, incorporating excessively large receptive fields can degrade performance due to detailed information loss.

## H  FAILURE CASE

Although MPAMA demonstrates promising performance across a wide range of objects, it is crucial to examine its failure cases to gain deeper insights into its behavior and limitations. To this end, we present visualizations of false detections in Figure 9. In the 'zipper' category, the regions highlighted with green squares indicate surface flaws that are not annotated as anomalies in the ground truth. A similar phenomenon is observed with subtle fabric defects near the zipper, which are visually abnormal yet unlabeled. Furthermore, MPAMA occasionally misinterprets background artifacts as anomalies. The third column displays several capsules placed on a desk with a visible black stain. Although the stain may appear anomalous, the annotation only focuses exclusively on object anomalies. This raises an important research question: How can we guide MPAMA to focus specifically on object-centric anomalies while disregarding irrelevant background artifacts?

## I  VISUALIZATION ON CATEGORY-WISE SEGMENTATION

Category-wise visualizations are provided to facilitate an intuitive understanding of MPAMA's detection performance.

Table 9: Subset-level performance comparison (AUROC) for anomaly segmentation on MVTec AD.

| Object name | WinCLIP | VAND | AnomalyCLIP | MPL | MPAMA |
|---|---|---|---|---|---|
| Carpet | 95.4 | 98.4 | 98.8 | 99.2 | 99.5 |
| Bottle | 89.5 | 83.4 | 90.4 | 91.9 | 93.5 |
| Hazelnut | 94.3 | 96.1 | 97.1 | 97.3 | 97.2 |
| Leather | 96.7 | 99.1 | 98.6 | 99.0 | 99.2 |
| Cable | 77.0 | 72.3 | 78.9 | 82.7 | 84.7 |
| Capsule | 86.9 | 92.0 | 95.8 | 97.1 | 97.1 |
| Grid | 82.2 | 95.8 | 97.3 | 98.2 | 98.2 |
| Pill | 80.0 | 76.2 | 92.0 | 93.1 | 93.9 |
| Transistor | 74.7 | 62.4 | 71.0 | 71.3 | 74.0 |
| Metal_nut | 61.0 | 65.4 | 74.4 | 72.0 | 75.2 |
| Screw | 89.6 | 97.8 | 97.5 | 97.4 | 98.1 |
| Toothbrush | 86.9 | 95.8 | 91.9 | 94.7 | 96.1 |
| Zipper | 91.6 | 91.1 | 91.4 | 96.7 | 97.9 |
| Tile | 77.6 | 92.7 | 94.6 | 97.7 | 97.1 |
| Wood | 93.4 | 95.8 | 96.5 | 98.0 | 98.0 |
| Mean | 85.1 | 87.6 | 91.1 | 92.5 | 93.2 |

Table 10: Subset-level performance comparison (PRO) for anomaly segmentation on MVTec AD.

| Object name | WinCLIP | VAND | AnomalyCLIP | MPL | MPAMA |
|---|---|---|---|---|---|
| Carpet | 84.1 | 48.5 | 90.1 | 97.5 | 96.4 |
| Bottle | 76.4 | 45.6 | 80.9 | 84.3 | 85.6 |
| Hazelnut | 81.6 | 70.3 | 92.4 | 91.1 | 92.1 |
| Leather | 91.1 | 72.4 | 92.2 | 98.0 | 98.0 |
| Cable | 42.9 | 25.7 | 64.4 | 74.7 | 78.5 |
| Capsule | 62.1 | 51.3 | 87.2 | 93.5 | 94.8 |
| Grid | 57.0 | 31.6 | 75.6 | 93.0 | 91.5 |
| Pill | 65.0 | 65.4 | 88.2 | 93.2 | 94.5 |
| Transistor | 43.4 | 21.3 | 58.1 | 55.6 | 57.1 |
| Metal_nut | 31.8 | 38.4 | 71.0 | 72.7 | 76.9 |
| Screw | 68.5 | 67.1 | 88.0 | 89.1 | 89.9 |
| Toothbrush | 67.7 | 54.5 | 88.5 | 92.1 | 92.4 |
| Zipper | 71.7 | 10.7 | 65.3 | 89.0 | 91.6 |
| Tile | 51.2 | 26.7 | 87.6 | 94.3 | 90.1 |
| Wood | 74.1 | 31.1 | 91.2 | 94.9 | 94.7 |
| Mean | 64.6 | 44.0 | 81.4 | 87.5 | 88.3 |

Table 11: Subset-level performance comparison (AUROC) for anomaly classification on MVTec AD.

| Object name | WinCLIP | VAND | AnomalyCLIP | MPL | MPAMA |
|---|---|---|---|---|---|
| Carpet | 100.0 | 99.5 | 100.0 | 99.7 | 100.0 |
| Bottle | 99.2 | 92.0 | 89.3 | 92.5 | 92.9 |
| Hazelnut | 93.9 | 89.6 | 97.2 | 94.6 | 94.5 |
| Leather | 100.0 | 99.7 | 99.8 | 99.9 | 99.7 |
| Cable | 86.5 | 88.4 | 69.8 | 94.3 | 94.4 |
| Capsule | 72.9 | 79.9 | 89.9 | 95.6 | 96.5 |
| Grid | 98.8 | 86.3 | 97.0 | 98.9 | 99.3 |
| Pill | 79.1 | 80.5 | 81.8 | 88.4 | 87.6 |
| Transistor | 88.0 | 80.8 | 92.8 | 89.4 | 89.5 |
| Metal_nut | 97.1 | 68.4 | 93.6 | 81.7 | 85.6 |
| Screw | 83.3 | 84.9 | 81.1 | 84.0 | 84.5 |
| Toothbrush | 88.0 | 53.8 | 84.7 | 96.8 | 96.8 |
| Zipper | 91.5 | 89.6 | 98.5 | 99.7 | 99.8 |
| Tile | 100.0 | 99.9 | 100.0 | 98.7 | 98.7 |
| Wood | 99.4 | 99.0 | 96.8 | 98.1 | 97.8 |
| Mean | 91.8 | 86.1 | 91.5 | 94.1 | 94.5 |

Table 12: Subset-level performance comparison (AP) for anomaly classification on MVTec AD.

| Object name | WinCLIP | VAND | AnomalyCLIP | MPL | MPAMA |
|---|---|---|---|---|---|
| Carpet | 100.0 | 99.8 | 100.0 | 99.9 | 100.0 |
| Bottle | 99.8 | 97.7 | 97.0 | 97.7 | 98.0 |
| Hazelnut | 96.9 | 94.8 | 98.6 | 97.2 | 96.9 |
| Leather | 100.0 | 99.9 | 99.9 | 99.9 | 99.9 |
| Cable | 91.2 | 93.1 | 81.4 | 96.7 | 96.4 |
| Capsule | 91.5 | 95.5 | 97.9 | 99.0 | 99.4 |
| Grid | 99.6 | 94.9 | 99.1 | 99.7 | 99.5 |
| Pill | 95.7 | 96.0 | 95.4 | 97.5 | 97.4 |
| Transistor | 87.1 | 77.5 | 90.6 | 88.4 | 88.9 |
| Metal_nut | 99.3 | 91.9 | 98.5 | 95.7 | 96.6 |
| Screw | 93.1 | 93.6 | 92.5 | 93.5 | 94.0 |
| Toothbrush | 95.6 | 71.5 | 93.7 | 98.6 | 98.6 |
| Zipper | 97.5 | 97.1 | 99.6 | 99.8 | 99.6 |
| Tile | 100.0 | 100.0 | 100.0 | 99.6 | 99.5 |
| Wood | 99.8 | 99.7 | 99.2 | 99.4 | 99.3 |
| Mean | 96.5 | 93.5 | 96.2 | 96.9 | 97.6 |

Table 13: Subset-level performance comparison (AUROC) for anomaly segmentation on VisA.

| Object name | WinCLIP | VAND | AnomalyCLIP | MPL | MPAMA |
|---|---|---|---|---|---|
| Candle | 88.9 | 97.8 | 98.8 | 98.5 | 98.6 |
| Capsules | 81.6 | 97.5 | 95.0 | 94.6 | 95.8 |
| Cashew | 84.7 | 86.0 | 93.8 | 95.3 | 95.5 |
| Chewinggum | 93.3 | 99.5 | 99.3 | 99.1 | 99.6 |
| Fryum | 88.5 | 92.0 | 94.6 | 94.8 | 95.0 |
| Macaroni1 | 70.9 | 98.8 | 98.3 | 98.6 | 99.0 |
| Macaroni2 | 59.3 | 97.8 | 97.6 | 97.4 | 97.6 |
| Pcb1 | 61.2 | 92.7 | 94.1 | 95.0 | 96.7 |
| Pcb2 | 71.6 | 89.7 | 92.4 | 92.9 | 92.8 |
| Pcb3 | 85.3 | 88.4 | 88.4 | 88.6 | 87.7 |
| Pcb4 | 94.4 | 94.6 | 95.7 | 96.7 | 96.7 |
| Pipe_fryum | 75.4 | 96.0 | 98.2 | 98.4 | 98.4 |
| Mean | 79.6 | 94.2 | 95.5 | 95.8 | 96.1 |

Table 14: Subset-level performance comparison (PRO) for anomaly segmentation on VisA.

| Object name | WinCLIP | VAND | AnomalyCLIP | MPL | MPAMA |
|---|---|---|---|---|---|
| Candle | 83.5 | 92.5 | 96.2 | 95.1 | 95.3 |
| Capsules | 35.3 | 86.7 | 78.5 | 80.2 | 84.9 |
| Cashew | 76.4 | 91.7 | 91.6 | 92.3 | 93.3 |
| Chewinggum | 70.4 | 87.3 | 91.2 | 92.7 | 90.7 |
| Fryum | 77.4 | 89.7 | 86.8 | 83.9 | 85.3 |
| Macaroni1 | 34.3 | 93.2 | 89.8 | 93.6 | 94.4 |
| Macaroni2 | 21.4 | 82.3 | 84.2 | 87.3 | 85.7 |
| Pcb1 | 26.3 | 87.5 | 81.7 | 83.7 | 88.2 |
| Pcb2 | 37.2 | 75.6 | 78.9 | 81.3 | 81.8 |
| Pcb3 | 56.1 | 77.8 | 77.1 | 76.4 | 76.6 |
| Pcb4 | 80.4 | 86.8 | 91.3 | 91.6 | 91.5 |
| Pipe_fryum | 82.3 | 90.9 | 96.8 | 96.5 | 95.4 |
| Mean | 56.8 | 86.8 | 87.0 | 87.9 | 88.6 |

Table 15: Subset-level performance comparison (AUROC) for anomaly classification on VisA.

| Object name | WinCLIP | VAND | AnomalyCLIP | MPL | MPAMA |
|---|---|---|---|---|---|
| Candle | 95.4 | 83.8 | 79.3 | 85.2 | 83.5 |
| Capsules | 85.0 | 61.2 | 81.5 | 90.5 | 89.8 |
| Cashew | 92.1 | 87.3 | 76.3 | 90.4 | 90.8 |
| Chewinggum | 96.5 | 96.4 | 97.4 | 97.6 | 97.2 |
| Fryum | 80.3 | 94.3 | 93.0 | 92.2 | 92.7 |
| Macaroni1 | 76.2 | 71.6 | 87.2 | 86.2 | 87.4 |
| Macaroni2 | 63.7 | 64.6 | 73.4 | 75.2 | 76.3 |
| Pcb1 | 73.6 | 53.4 | 85.4 | 75.9 | 76.4 |
| Pcb2 | 51.2 | 71.8 | 62.2 | 66.0 | 68.8 |
| Pcb3 | 73.4 | 66.8 | 62.7 | 66.7 | 68.2 |
| Pcb4 | 79.6 | 95.0 | 93.9 | 97.8 | 98.0 |
| Pipe_fryum | 69.7 | 89.9 | 92.4 | 96.0 | 96.2 |
| Mean | 78.1 | 78.0 | 82.1 | 85.0 | 85.4 |

Table 16: Subset-level performance comparison (AP) for anomaly classification on VisA.

| Object name | WinCLIP | VAND | AnomalyCLIP | MPL | MPAMA |
|---|---|---|---|---|---|
| Candle | 95.8 | 86.9 | 81.1 | 87.9 | 86.4 |
| Capsules | 90.9 | 74.3 | 88.7 | 94.8 | 94.7 |
| Cashew | 96.4 | 94.1 | 89.4 | 95.7 | 95.7 |
| Chewinggum | 98.6 | 98.4 | 98.9 | 98.8 | 99.0 |
| Fryum | 90.1 | 97.2 | 96.8 | 96.1 | 96.5 |
| Macaroni1 | 75.8 | 70.9 | 86.0 | 87.1 | 88.4 |
| Macaroni2 | 60.3 | 63.2 | 72.1 | 74.9 | 76.6 |
| Pcb1 | 78.4 | 57.2 | 87.0 | 78.8 | 78.6 |
| Pcb2 | 49.2 | 73.8 | 64.3 | 68.5 | 71.1 |
| Pcb3 | 76.5 | 70.7 | 70.0 | 73.1 | 73.1 |
| Pcb4 | 77.7 | 95.1 | 94.4 | 97.4 | 97.7 |
| Pipe_fryum | 82.3 | 94.8 | 96.3 | 97.8 | 98.5 |
| Mean | 81.2 | 81.4 | 85.4 | 87.6 | 88.0 |

Table 17: Subset-level performance comparison (AUROC) for anomaly segmentation on MPDD.

| Object name | WinCLIP | VAND | AnomalyCLIP | MPL | MPAMA |
|---|---|---|---|---|---|
| Bracket_black | 57.8 | 96.3 | 95.7 | 96.4 | 97.0 |
| Bracket_brown | 72.2 | 86.2 | 94.4 | 94.5 | 95.2 |
| Bracket_white | 79.5 | 99.0 | 99.8 | 99.4 | 100.0 |
| Connector | 79.0 | 90.6 | 97.2 | 96.9 | 97.3 |
| Metal_plate | 92.6 | 93.1 | 93.8 | 94.7 | 93.6 |
| Tubes | 77.6 | 99.1 | 98.1 | 96.9 | 97.6 |
| Mean | 76.4 | 94.1 | 96.5 | 96.5 | 96.8 |

Table 18: Subset-level performance comparison (PRO) for anomaly segmentation on MPDD.

| Object name | WinCLIP | VAND | AnomalyCLIP | MPL | MPAMA |
|---|---|---|---|---|---|
| Bracket_black | 43.0 | 89.7 | 85.2 | 87.3 | 88.9 |
| Bracket_brown | 25.0 | 70.3 | 77.7 | 77.4 | 81.3 |
| Bracket_white | 57.6 | 93.1 | 98.8 | 98.1 | 98.8 |
| Connector | 44.6 | 74.5 | 89.8 | 89.3 | 88.4 |
| Metal_plate | 78.2 | 74.5 | 86.9 | 88.8 | 86.9 |
| Tubes | 44.7 | 96.9 | 93.6 | 88.9 | 90.4 |
| Mean | 48.9 | 83.2 | 88.7 | 88.3 | 89.2 |

Table 19: Subset-level performance comparison (AUROC) for anomaly classification on MPDD.

| Object name | WinCLIP | VAND | AnomalyCLIP | MPL | MPAMA |
|---|---|---|---|---|---|
| Bracket_black | 41.5 | 66.1 | 67.3 | 68.6 | 71.1 |
| Bracket_brown | 48.6 | 64.0 | 62.2 | 61.6 | 63.0 |
| Bracket_white | 40.2 | 79.6 | 64.9 | 78.7 | 80.7 |
| Connector | 79.3 | 78.8 | 86.9 | 82.4 | 81.5 |
| Metal_plate | 93.4 | 53.8 | 85.2 | 88.5 | 91.8 |
| Tubes | 78.7 | 95.9 | 95.5 | 96.5 | 99.2 |
| Mean | 63.6 | 73.0 | 77.0 | 79.4 | 81.2 |

Table 20: Subset-level performance comparison (AP) for anomaly classification on MPDD.

| Object name | WinCLIP | VAND | AnomalyCLIP | MPA | MPAMA |
|---|---|---|---|---|---|
| Bracket_black | 56.9 | 71.7 | 72.9 | 75.7 | 75.7 |
| Bracket_brown | 69.5 | 79.0 | 80.8 | 76.5 | 78.5 |
| Bracket_white | 45.1 | 82.3 | 68.5 | 76.4 | 76.3 |
| Connector | 61.3 | 71.8 | 76.8 | 67.0 | 65.1 |
| Metal_plate | 97.6 | 78.3 | 94.7 | 94.9 | 97.3 |
| Tubes | 89.1 | 98.1 | 98.1 | 98.0 | 99.5 |
| Mean | 69.9 | 80.2 | 82.0 | 81.4 | 82.1 |

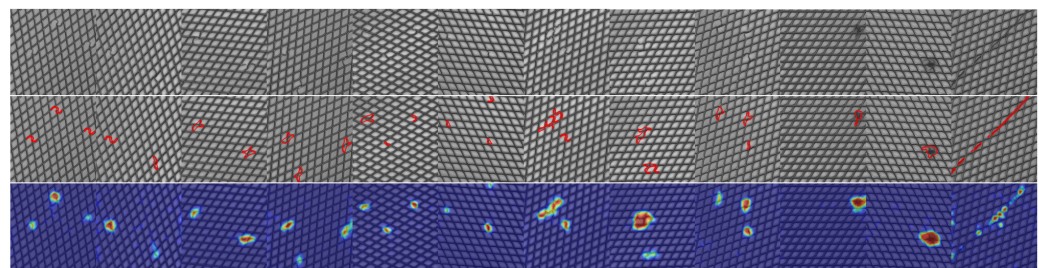

Figure 10: Anomaly segmentation for the 'grid' subset of MVTec AD. The first row shows the input images, the second row highlights anomaly regions with red contours, and the third row displays the segmentation results from MPAMA.

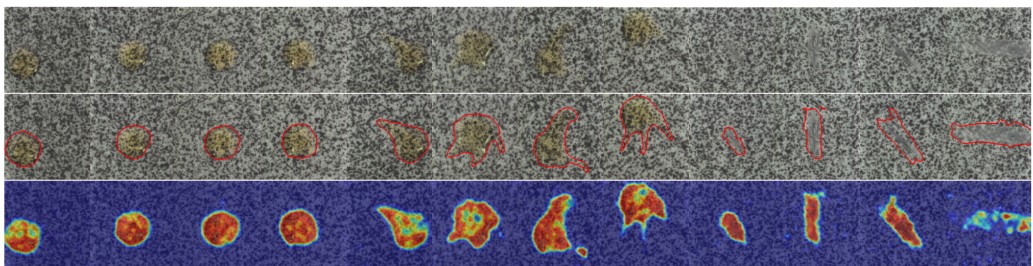

Figure 11: Anomaly segmentation for the 'skin' subset of ISIC. The first row shows the input images, the second row highlights anomaly regions with red contours, and the third row displays the segmentation results from MPAMA.

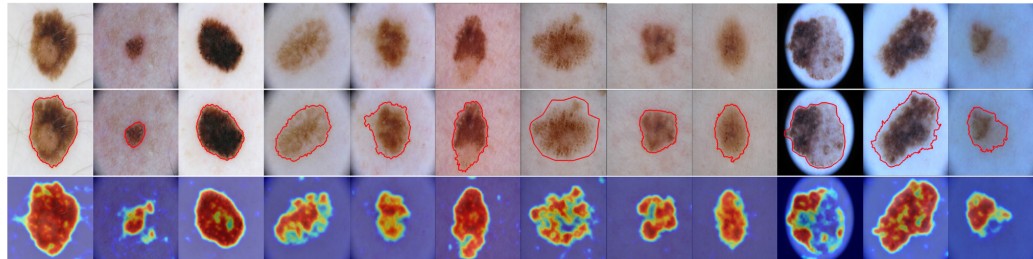

Figure 12: Anomaly segmentation for the 'tile' subset of MVTec AD. The first row shows the input images, the second row highlights anomaly regions with red contours, and the third row displays the segmentation results from MPAMA.

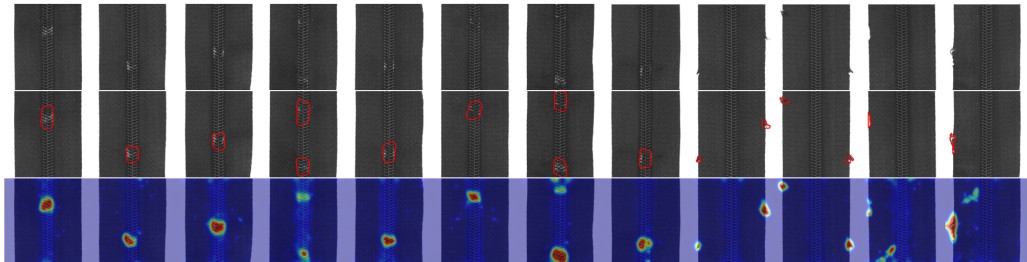

Figure 13: Anomaly segmentation for the 'zipper' subset of MVTec AD. The first row shows the input images, the second row highlights anomaly regions with red contours, and the third row displays the segmentation results from MPAMA.

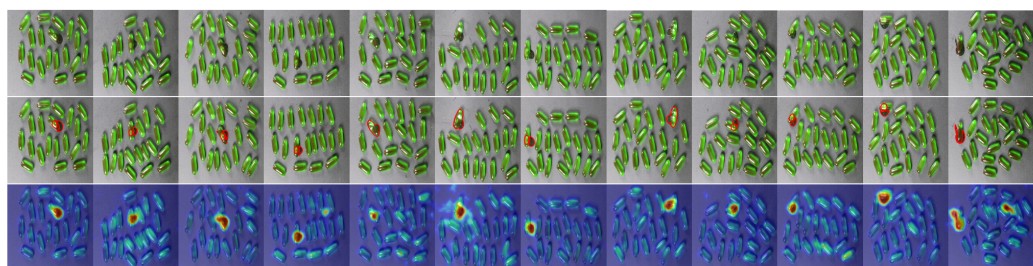

Figure 14: Anomaly segmentation for the 'capsule' subset of VisA. The first row shows the input images, the second row highlights anomaly regions with red contours, and the third row displays the segmentation results from MPAMA.

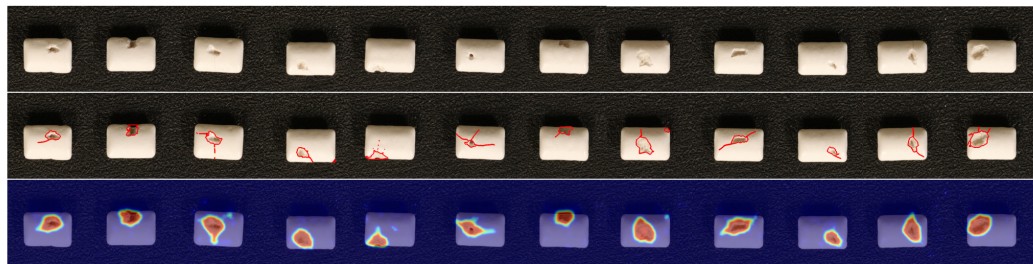

Figure 15: Anomaly segmentation for the 'chewinggum' subset of VisA. The first row shows the input images, the second row highlights anomaly regions with red contours, and the third row displays the segmentation results from MPAMA.

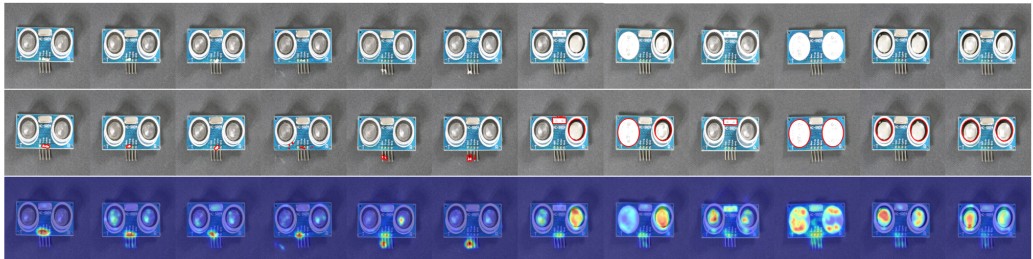

Figure 16: Anomaly segmentation for the 'pcb1' subset of VisA. The first row shows the input images, the second row highlights anomaly regions with red contours, and the third row displays the segmentation results from MPAMA.

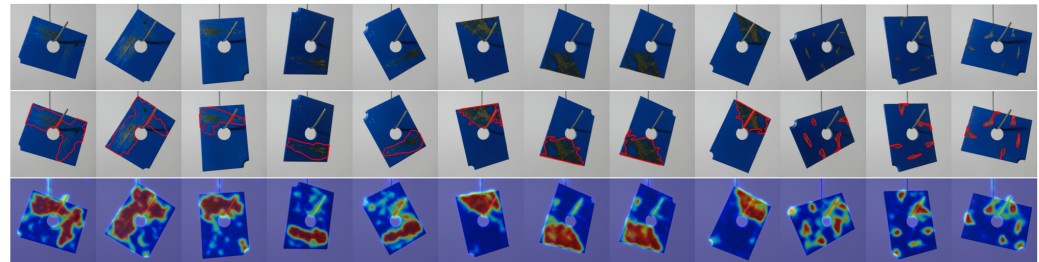

Figure 17: Anomaly segmentation for the 'meta-plate' subset of VisA. The first row shows the input images, the second row highlights anomaly regions with red contours, and the third row displays the segmentation results from MPAMA.

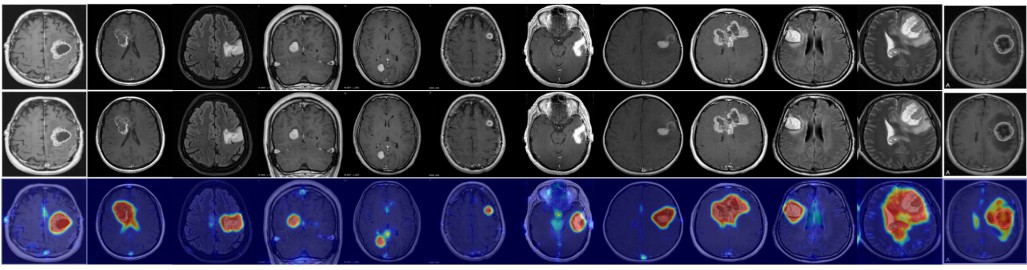

Figure 18: Anomaly segmentation for the 'brain' subset of HeadCT. The first row shows the input images, the second row highlights anomaly regions with red contours, and the third row displays the segmentation results from MPAMA.

