# OpenReview forum: "Learning Multi-granularity Visual-textual Alignment for Zero-shot Anomaly Detection"
_ICLR.cc/2026/Conference — Submitted to ICLR 2026_

### Official Review · Reviewer_Wx5L · 2025-10-30

**Soundness:** 4
**Presentation:** 3
**Contribution:** 3
**Rating:** 4
**Confidence:** 5

**Summary:**

This paper proposes MPAMA, a framework for zero-shot anomaly detection (ZSAD). It argues that existing CLIP-based methods suffer from single-level visual-textual alignment. To address this, MPAMA introduces a two-stage approach: 1) Multi-Perception Alignment (MPA), which uses text prompts with different receptive fields, and 2) Mixture-of-Experts Adaptation (MA), which routes visual tokens to specialized experts. The authors claim state-of-the-art performance on multiple industrial and medical datasets.

**Strengths:**

1. The proposed Multi-Perception Alignment (MPA) module directly addresses the challenge of variable anomaly sizes through a structured, learnable prompt design.

2. The paper provides extensive evaluations across 15 datasets from industrial and medical domains, demonstrating a serious effort to prove generalizability.

3. The authors include ablation studies to isolate the contributions of the main components (MPA, MA), which aids in understanding the proposed architecture.

**Weaknesses:**

1. The core idea of multi-scale perception for anomaly detection in CLIP is not novel. Specifically, the method MUSC[1] has already effectively demonstrated multi-scale cropping and feature aggregation for zero-shot anomaly localization with superior performance. While MPAMA uses prompt learning instead of visual crops, the fundamental multi-scale principle is similar. The paper's failure to cite, discuss, or quantitatively compare against MUSC or similar multi-scale CLIP adaptations significantly overstates its novelty.

2. The entire experimental section is based exclusively on the CLIP ViT-L/14 backbone. The performance and the proposed architectural contributions (especially the MoE module) are not validated on other CLIP variants (e.g., ViT-B/16, ResNet-based backbones). This lack of backbone-agnostic validation raises serious doubts about the generalizability and fundamental robustness of the proposed method.


3. The expert selection frequency (Figure 5/Table 5) shows a severe load imbalance, with one expert handling ~50% of the tokens. The "zero expert" is used for only 0.4% of tokens, rendering it functionally redundant. The authors provide no ablation study to demonstrate its necessity, making it seem like an unjustified and ineffective component that adds unnecessary complexity.

4. The paper fails to compare its method with AA-CLIP, a very recent and relevant SOTA method. Furthermore, a comprehensive comparison must include not just accuracy but also critical efficiency metrics like inference speed (FPS) and GPU memory against all key baselines. The provided overhead analysis is insufficient without direct comparison with other methods, hiding potential inefficiencies.

5. Typos: (1) Eq.3: CrossEntroy - CrossEntropy (2) Line 348-349: These illustrated - These results illustrate/This illustrates

[1] Li X, Huang Z, Xue F, et al. Musc: Zero-shot industrial anomaly classification and segmentation with mutual scoring of the unlabeled images[C]//The Twelfth International Conference on Learning Representations. 2024.

**Questions:**

Please addressing these points, particularly by providing the requested comparisons with MUSC and AA-CLIP, validating the method on another backbone, and clarifying the MoE design choices.

---

### Official Review · Reviewer_EKC5 · 2025-10-30

**Soundness:** 2
**Presentation:** 2
**Contribution:** 2
**Rating:** 4
**Confidence:** 5

**Summary:**

This paper introduces MPAMA, a novel framework for zero-shot anomaly detection designed to learn multi-granularity representations. The authors argue that existing methods are limited by single-level visual-textual alignment, which fails to capture the diverse scales and appearances of anomalies. Their solution is a two-stage process. The first stage, MPA, adapts the textual space by introducing multi-perception prompts that align with visual features aggregated at various spatial scales. It also disentangles prompts for local and global semantics. The second stage, MA, freezes the textual space and adapts the visual space using a MoE architecture. This module dynamically routes visual patch tokens to a pool of specialized experts, allowing for a more nuanced, token-wise visual representation. The authors validate their approach with extensive experiments on numerous industrial and medical datasets, demonstrating superior performance over current state-of-the-art methods.

**Strengths:**

1. The paper provides strong and comprehensive empirical evidence. The proposed models, MPA and MPAMA, consistently outperform strong baselines across 15 datasets from different domains, which robustly demonstrates the effectiveness of the proposed techniques.

**Weaknesses:**

1. The paper frames this as enriching the textual space by creating prompts with different receptive fields. However, the described mechanism appears to use standard text embeddings and compare them against visual features that have been aggregated over different patch neighborhoods. This suggests the multi-scale awareness is primarily introduced on the visual side of the similarity computation, not within the textual prompts themselves.

2. The rationale behind the strict two-stage training process is not fully elaborated. This is a significant design choice that likely impacts the final representation. The paper does not explore or justify why a joint, end-to-end optimization of both MPA and MA modules is not employed, which leaves questions about the stability and dynamics of the learning process.

3. The framework's overall complexity is high, involving multiple perception levels, disentangled prompts, a cross-bridge mechanism, a two-stage training procedure, and a Mixture-of-Experts module.

**Questions:**

1. The paper states it enriches the textual space with prompts of diverse receptive fields. However, the mechanism seems to involve comparing standard text embeddings to visually aggregated features. Could the authors clarify if the text prompts themselves have any inherent scale information, or if the multi-scale nature is purely a function of the visual feature processing prior to the alignment step?

2. The framework adapts the text space first and then the visual space. What is the empirical or theoretical justification for this specific order? Have the authors experimented with reversing the stages or with a joint end-to-end training approach?

3. The MoE module includes a "zero expert" for tokens that are presumably easy to classify. How is this implemented, and what is its functional advantage over a standard sparse routing mechanism where the router could simply learn to assign near-zero weights to all available experts for a given token?

---

### Official Review · Reviewer_QvaM · 2025-10-31

**Soundness:** 2
**Presentation:** 2
**Contribution:** 2
**Rating:** 4
**Confidence:** 5

**Summary:**

This paper presents a framework for ZSAD that aims to improve performance by learning multi-granularity visual-textual representations. The authors identify two primary limitations in existing methods: (1) single-level alignment, which struggles to capture anomalies of varying spatial scales, and (2) lack of visual granularity, where a single visual expert is insufficient to model diverse anomaly patterns. To address these, the paper proposes a two-stage approach. The first stage, MPA, adapts the textual space by introducing "multi-perception text prompts," which are designed with different receptive fields to capture anomalies at multiple scales. These prompts are created by aggregating visual features over different neighborhood sizes. The second stage, MA, freezes the text space and adapts the visual space. It replaces the single visual expert with a pool of specialized experts and uses a dynamic routing mechanism to assign different visual tokens to different experts. The complete framework is named MPAMA. The method's effectiveness is demonstrated through extensive experiments on industrial and medical datasets.

**Strengths:**

1.The proposed method, particularly the full MPAMA framework, achieves state-of-the-art performance on a wide range of industrial and medical benchmarks for both image-level and pixel-level anomaly detection tasks.

2.The idea of explicitly modeling multi-scale information is well-motivated, as real-world anomalies often vary significantly in size. The introduction of multi-perception prompts and a Mixture-of-Experts architecture are logical extensions to existing ZSAD frameworks.

**Weaknesses:**

1.The work appears to be an incremental development over existing ZSAD methods rather than a solution to a clearly defined, unsolved problem. The paper does not specify what types of anomalies are undetectable by current methods but can be successfully identified by the proposed multi-granularity approach. The motivation is framed in general terms (anomalies vary in shape and appearance), but it lacks concrete examples or analysis of failure cases in prior work that the proposed method specifically overcomes.

2.The functional roles of the different multi-perception text prompts are not experimentally validated. The framework introduces a set of prompts corresponding to different receptive fields, but there is no analysis to demonstrate that each prompt specializes in detecting anomalies of a particular scale. For example, it is not shown whether the prompt with the smallest receptive field is uniquely responsible for detecting "tiny" anomalies or if the prompt with the largest field is critical for "large" ones. Without such evidence, it is difficult to confirm that the multi-scale mechanism is functioning as intended.

3.The paper lacks a clear explanation of the mechanism that drives different experts in the MoE module to specialize in different anomaly types. What principles or inductive biases in the training process encourage one expert to focus on "background and texture" while another captures "deformations"?

**Questions:**

1.Could the authors provide an analysis to demonstrate a direct correlation between a prompt of a certain perception level (e.g., p=1) and the model's ability to detect anomalies of a corresponding scale (e.g., "tiny")? For instance, would ablating the p=1 prompt lead to a significant performance drop specifically on tiny anomalies, while leaving performance on large anomalies largely unaffected?

2.The proposed framework involves a two-stage training process where the textual space is adapted first, followed by the visual space. What is the rationale for this sequential approach instead of joint end-to-end training? Have the authors experimented with a joint training scheme, and if so, how did it compare to the two-stage process? Does the sequential training help stabilize the learning process or prevent interference between the textual and visual adaptation modules?

3.The MoE module is shown to route visual tokens to different experts that purportedly specialize in distinct visual patterns. What is the underlying mechanism or inductive bias that drives this specialization during training? Is it purely an emergent property of the routing optimization, or are there explicit constraints or architectural designs that encourage this division of labor? A more in-depth explanation of the learning dynamics that lead to semantic specialization is needed.

---

### Official Review · Reviewer_uoNW · 2025-10-31

**Soundness:** 2
**Presentation:** 3
**Contribution:** 2
**Rating:** 4
**Confidence:** 4

**Summary:**

This paper proposes MPAMA, a framework for ZSAD that enhances CLIP’s visual-textual alignment from single-scale to multi-granularity. MPAMA demonstrates strong results across seven industrial and eight medical datasets, achieving new SOTA on diverse datasets. Ablations confirm that both multi-scale textual prompts and expert routing contribute significantly.

**Strengths:**

1.The paper moves beyond existing single-level or ensemble prompt schemes by explicitly assigning receptive fields to text prompts and integrating a cross-level bridge between local and global semantics.

2.The visual-expert routing provides clear specialization (frozen/learnable/zero experts) and interpretable routing frequencies (Fig. 5). It offers a balanced compromise between generalization and specialization.

3.Experiments cover diverse industrial and medical datasets, with consistent SOTA gains and insightful ablations on perception levels and expert counts, reinforcing the framework’s robustness.

**Weaknesses:**

1.The paper does not use the officially reported results of FAPrompt (ICCV 2025) and without comparison with FiLo (MM 2024), which is a major concurrent fine-grained prompt-based method.

2.The paper claims that different experts in the MoE adaptation specialize in diverse visual patterns (e.g., background, texture, deformation). However, the provided evidence (Fig. 1d and routing frequency histogram in Fig. 5) remains largely qualitative and dataset-level. It is unclear whether the experts truly learn complementary or distinct feature subspaces rather than redundant representations.

3.Although the proposed MPAMA achieves new SOTA on several datasets, the absolute gains over prior methods (e.g., AnomalyCLIP, FAPrompt) are relatively small in most of datasets.

**Questions:**

1.Have you visualized attention maps per expert to confirm semantic specialization?

2.How critical is the max-pool fusion between local and global prompts? could other fusion (attention, gating) yield better transfer?

---

### Meta-Review · Area_Chair_xUGd · 2026-01-06

**Summary:**

Concerns include methodological validation gaps, insufficient comparisons to related work, unclear design rationales, and limited novelty/evidence for key components.

**Reviewer Concerns:**

no rebuttal

**Reviewer Scores:**

no discussion

---

### Decision · Program_Chairs · 2026-01-26

Reject